# Factors associated with in-hospital mortality of patients admitted to an intensive care unit in a tertiary hospital in Malawi

**Mtisunge Kachingwe**[1,2]*, **Raphael Kazidule Kayambankadzanja**[1,2], **Wezzie Kumwenda Mwafulirwa**[1], **Singatiya Stella Chikumbanje**[1,2], **Tim Baker**[1,2,3,4]

**1** Department of Anaesthesia and Intensive Care, Queen Elizabeth Central Hospital, Blantyre, Malawi, **2** Kamuzu University of Health Sciences, Blantyre, Malawi, **3** Department of Global Public Health, Karolinska Institutet, Stockholm, Sweden, **4** Perioperative Medicine and Intensive Care, Karolinska University Hospital, Stockholm, Sweden

* mtkachingwe@gmail.com

## Abstract

### Objective

To determine factors associated with in-hospital death among patients admitted to ICU and to evaluate the predictive values of single severely deranged vital signs and several severity scoring systems.

### Methods

A combined retrospective and prospective cohort study of patients admitted to the adult ICU in a tertiary hospital in Malawi was conducted between January 2017 and July 2019. Predefined potential risk factors for in-hospital death were studied with univariable and multivariable logistic regression models, and the performance of severity scores was assessed.

### Results

The median age of the 822 participants was 31 years (IQR 21–43), and 50% were female. Several factors at admission were associated with in-hospital mortality: the presence of one or more severely deranged vital signs, adjusted odds ratio (aOR) 1.9 (1.4–2.6); treatment with vasopressor aOR 2.3 (1.6–3.4); received cardiopulmonary resuscitation aOR 1.7 (1.2–2.6) and treatment with mechanical ventilation aOR 1.5 (1.1–2.1). Having had surgery had a negative association with in-hospital mortality aOR 0.5 (0.4–0.7). The predictive accuracy of the severity scoring systems had varying sensitivities and specificities, but none were sufficiently accurate to be clinically useful.

### Conclusions

In conclusion, the presence of one or more severely deranged vital sign in patients admitted to ICU may be useful as a simple marker of an increased risk of in-hospital death.

**Data Availability Statement:** The Authors share the belief that sharing data fosters scientific progress. In this case the data primarily belongs to

the department of Anaesthesia at Queen Elizabeth Central Hospital. We the researchers had to seek permission from the College of Medicine Research and Ethics Committee (Reference number (P.07/18/2433)) and Department of Anaesthesia and Intensive care (QECH, Malawi) to access the database and at that time permission did not extend to sharing it publicly. Data is accessible upon request. Please direct all data access queries to Dr. Samson Mndolo, Hospital director Queen Elizabeth Central Hospital, email: samson.mndolo@mail.gov.mw. The department would like to review each request on a case by case basis. We the authors can confirm that we had no special access privileges that allowed us access to this data."

**Funding:** RKK and TB received grants for the study from Life Support Foundation (https://www.lifesupportfoundation.org/) and The Association of Anaesthetists (https://anaesthetists.org/) The funders had no role in study design, data collection and analysis, decision to publish, or preparation of the manuscript.

**Competing interests:** The authors have declared that no competing interests exist.

## Introduction

Intensive care in low income countries (LICs) is at an early stage of development despite the high burden of critical illness [1–3]. Good quality critical care provided in intensive care units (ICUs) has the potential to reduce mortality and morbidity among critically ill patients [4].

Illness severity is a major determinant of patient outcomes in an ICU: the sicker the patient at admission the greater the risk of a poor outcome. A patient's vital signs (heart rate, respiratory rate, blood pressure, conscious level, oxygen saturation) are commonly used as markers of illness severity in hospitals and are components of compound scoring systems [5–7]. Deranged vital signs have shown to correlate with negative outcomes such as cardiac arrest and mortality [8, 9].

Data about intensive care services in LICs are limited. In Malawi, there are gaps in knowledge about patient characteristics, illness severity on admission to ICU and the risk factors for poor outcomes. Understanding the factors associated with poor outcomes is a vital step for understanding which patients are most likely to benefit from the limited ICU capacity and to direct treatment protocols in the ICU. The primary aim of this study was to identify factors associated with in-hospital death among patients admitted to ICU. A secondary aim was to evaluate the ability of single severely deranged vital signs and other severity scoring systems to predict in-hospital mortality.

## Materials and methods

A combined retrospective and prospective cohort study was conducted of patients admitted to the adult ICU in Queen Elizabeth Central Hospital (QECH) in Blantyre, Malawi between January 2017 and July 2019.

### Setting

QECH is a state-run hospital in the southern region of Malawi with a bed capacity of 1200. It serves an immediate catchment population of 1 million and is a referral center receiving patients from across the country. The adult ICU has four beds and admits patients from all wards and all specialties in the hospital. The adult ICU provides mechanical ventilation, inotropic support and close monitoring and is staffed by an anesthesiologist, anesthetic clinical officers and nurses. On average there is a nurse-to-patient ratio of 1:1. QECH has an additional ICU for paediatric patients. Occasionally the adult ICU will admit paediatric patients when the paediatric ICU is full.

### Data collection

All patients admitted to the adult ICU during the study period were included as study participants. In December 2017, the department created an electronic database of all admissions, including data entered from unit's paper records for patients from 1st January to 30th November 2017 and prospectively directly from the patient from 1st December 2017. The prospective data were collected by the nurses immediately at admission or within 1 hour of admission if there were logistical delays in documentation, using a paper-based data collection tool. The data collection tool collected data on demographics, diagnosis, and ward from which they were admitted, vital signs, lab investigations and interventions received. The data extracted from the records for patients admitted between January and November included the first vital signs recorded during the first hour of admission to the ICU. Lab investigation results were obtained from the patients' files. Data collection was supervised for quality in the ICU and double-data entered into the database. Follow-up of all the patients continued on the wards

until hospital discharge or death. The primary endpoint for the study was in-hospital death. Patients lacking data on hospital outcome were excluded from analysis. The research team accessed the database at the end of July 2019.

## Data management

The variables of interest for the primary aim were predefined by the researchers as those that were clinically plausible to be associated with in-hospital mortality. They were 1) any single severely deranged vital sign (Fig 1); 2) treatment with an inotrope or vasopressor; 3) received cardiopulmonary resuscitation; 4) treated with mechanical ventilation; 5) positive HIV status; 6) delayed capillary refill time (>3seconds); 7) sex; 8) undergone surgery before admission to ICU; 9) age (categorised as <50 or ≥ 50 years); 10) nature of admission (planned or emergency); and 11) deranged temperature (<35.5˚C or > = 38.5 if aged <1 month and <35 or >40˚C if aged > = 1 month). Cardiopulmonary resuscitation comprised advanced life support (ALS) to patients who suffered a cardiac arrest. [10] Cardiac arrest was defined as, "the cessation of cardiac mechanical activity confirmed by the absence of a detectable pulse, unresponsiveness, and apnea (or agonal respirations)" [11]. The definition of "any single severely deranged vital sign" for adults was based on previous studies conducted in Tanzania [12–14] (Fig 1). For children, the definition of "any single severely deranged vital sign" were obtained from a study done in Malawi identifying risk factors for mortality in severely ill children [15]. The blood pressure definitions in children were was based on the paediatric early warning score [16].

## Data analysis

Data analysis was done with STATA (Release 14, StataCorp, College Station, TX). Descriptive data were summarized using proportions, means, ranges, medians and interquartile ranges

| Vital Sign | Age | Severely deranged |
| --- | --- | --- |
| Respiratory rate | < 1 month | <20 or >80/min |
| | 1 month - < 1 year | <15 or >60/min |
| | 1 year - <5 years | <10 or > 50/min |
| | 5 years – 12 years | <8 or >40/min |
| | >12 years | <8 or > 30/min |
| Saturation (%) | All | <90% |
| Pulse rate/minute | < 1 month | <80 or >200 bpm |
| | 1 month - < 1 year | <80 or > 180 bpm |
| | 1 year - <5 years | <70 or >170 bpm |
| | 5 years - 12 years | <60 or >150 bpm |
| | >12 years | <40 or >130 bpm |
| Glasgow coma score | All | </= 8/15 |
| Systolic blood pressure(mmhg) | < 3 months | <50 |
| | 3 months - < 1 year | <70 |
| | 1 year - < 4 years | <75 |
| | 4 years – < 12 years | <80 |
| | >=12 years | <90 |

Fig 1. Cut–offs for severely deranged vital signs (13, 14).

where appropriate. Diagnoses were categorised by the researchers into eight groups: 1) serious infection including endometritis, pneumonia, sepsis/septic shock, tuberculosis, malaria and meningitis; 2) non communicable diseases which included cancer, anaemia and unspecified tumours; 3) trauma which included head injury and burns; 4) bowel obstruction and perforation including typhoid perforation 5) acute respiratory disease including asthma and pulmonary oedema; 6) post-delivery and abortion care; 7) pre-eclampsia/eclampsia; 8) other/ unknown. Univariable and multivariable logistic regression including all the defined variables were used to generate odds ratios with a 95% confidence interval and significance level of <0.05. Missing data was handled using imputation. The primary method was imputing normal values for missing data with the assumption that normality is more common, and danger-signs are more likely to be documented. To test this assumption, two sensitivity analyses were conducted, firstly using complete case deletion when data were missing and secondly imputing missing data as danger-signs (S1 and S8 Tables). As an additional a-priori planned analysis, analyses of the associations of each individually deranged vital sign with in-hospital death were also conducted.

For the secondary aim, we compared predictive values of binary critical/non-critical scores of the severity scoring systems: Universal Vital signs Assessment (UVA), quick Sequential Organ Failure Assessment (qSOFA), National Early Warning Score (NEWS), Malawi Intensive care Mortality risk Evaluation model (MIME), Tropical Intensive Care Score (TROPICS), TOTAL score (Tachypnoea, Oxygen saturation, Temperature, Alert and Loss of independence) plus the presence of any single severely deranged vital sign [5, 7, 14, 17–20]. These models were chosen for their greater potential feasibility in low resourced settings than other models such as the Acute Physiology and Chronic Health Evaluation (APACHE), the Simplified Acute Physiology Score (SAPS) and the Mortality Probability Model (MPM). A description of the models is provided in S2 Table.

Binary cut-offs of the scoring systems into critical/non-critical scores were used for reasons of simplicity—in our opinion in low-resourced environments with few staff, calculating and using complex compound scores is not feasible. The cutoffs chosen for the critical/non-critical scores for each scoring system were defined in the original papers except for the MIME score, for which we set the cut-off. For the secondary analyses, only patients 16 years of age and above were included and each patient's qSOFA, UVA, NEWS, MIME and TROPICS scores were calculated from their component variables. Logistic regression analyses were done for each model with the outcome of in-hospital death. To assess the performance of the scoring systems for predicting death, sensitivity, specificity, positive predictive values (PPVs) and negative predictive values (NPVs) were calculated. Additional analyses were conducted with the patients stratified as medical or surgical and whether they were included in the retrospective or prospective data collection periods. Further exploratory analyses were done using different cut-offs and the overall performance of the scores assessed using the Area Under the Receiver Operator Curve (AUROCs).

Ethical clearance was granted by the Malawi College of Medicine Research and Ethics Committee (P.07/18/2433).

## Results and discussion

### Demographics

A total of 824 patients were admitted in the ICU during the study period. Data on hospital outcome was not found for two patients and thus 822 patients were included in the analysis. The median age of the participants was 31 years, (Inter Quartile Range (IQR) 21–43 years), and 408 were female (50%). Fifty-three percent of the patients were admitted from the operating

theatres, and the others were from the emergency departments, the hospital wards and from other hospitals. Patients were admitted from all specialties and were cared for in the ICU for a median of 2 days, (IQR 1–4 days) (Table 1). One hundred patients were under 16 years old—their median age was 5 years (IQR 1–10 years).

## Illness severity and outcomes

At admission, 525 patients (64%) had one or more severely deranged vital sign. The 24-hour ICU mortality for all the patients was 11% (87 patients), 284 patients died during their ICU stay (35%) and 368 patients died in-hospital (45%).

**Table 1. Patient characteristics.**

| Variable | n (%) N = 822 |
|---|---|
| Age Median (IQR) | 31 (21–43) |
| Female Sex | 408 (50) |
| Admitted to ICU from | |
| Operating theatres | 434(53) |
| Emergency department | 187 (23) |
| Hospital ward | 164(20) |
| Recovery room | 3 (0.4) |
| Other hospitals | 19(2) |
| Unknown | 15 (3) |
| Admission type | |
| Emergency | 713 (87) |
| Planned | 110(13) |
| Specialty | |
| Surgery* | 326 (40) |
| Medicine | 180 (22) |
| Neurosurgery | 132 (16) |
| Obstetrics and gynaecology | 135 (16) |
| Paediatrics | 49 (6) |
| Had surgery in hospital before admission to ICU | 515 (63) |
| Positive HIV status n/N (%)** | 32/114 (28) |
| Diagnosis | |
| Serious infection*** | 182 (22) |
| Non-Communicable disease**** | 178 (22) |
| Trauma***** | 135 (16) |
| Bowel perforation or obstruction****** | 121 (15) |
| Acute respiratory disease ******* | 47 (6) |
| Post-delivery or abortion care | 49 (6) |
| Pre-eclampsia/eclampsia | 20 (2) |
| Other/unknown | 90 (11) |

* includes Orthopedics, Plastics/Burns, ENT, ophthalmology

** N = Number of patients with known HIV status

***Including Meningitis, Malaria, Endometritis, Pneumonia, sepsis/septic shock and Tuberculosis

**** Cancer, Anaemia, unspecified tumours

*****Including Head injury and Burns

******including Typhoid perforation

******* includes Asthma and Pulmonary Oedema

## Factors associated with in-hospital mortality

Using univariable analysis, the following factors were associated with in-hospital mortality: the presence of one or more severely deranged vital sign (unadjusted Odds Ratio (uOR 2.5, 95% CI; 1.8–3.3)); treatment with inotrope or vasopressor (uOR 2.9, 95% CI; 2.0–4.0); received cardiopulmonary resuscitation (uOR 2.3, 95% CI; 1.6–3.3); treatment with mechanical ventilation (uOR 1.9, 95% CI; 1.4–2.5) and age >50 (uOR 1.5, 95% CI; 1.1–2.2). Having had surgery had a negative association with in-hospital mortality (uOR 0.6, 95% CI; 0.4–0.8) (Table 2). In the additional analysis of individual severely deranged vital signs, hypotension (uOR 2.0, 95% CI; 1.4–2.8), hypoxia (uOR 3.2, 95% CI 2.0–5.2) and low consciousness level (uOR 1.8, 95% CI; 1.3–2.4) were found to be associated with in-hospital mortality (S3 Table).

In multivariable analysis the same factors were associated with in-hospital mortality; the presence of one or more severely deranged vital sign (adjusted OR 1.9, 95% CI; 1.4–2.6); treatment with inotrope or vasopressor (aOR 2.3, 95% CI; 1.6–3.3); received cardiopulmonary resuscitation (aOR 1.7, 95% CI; 1.2–2.5)); and treatment with mechanical ventilation (aOR 1.5, 95% CI; 1.1–2.1). However the association of age > 50 (aOR 1.4, 95% CI; 1.0–2.1)) with in-hospital mortality was not statistically significant. Having had surgery had a negative association with in-hospital mortality (aOR 0.5, 95% CI; 0.4–0.7).

## Predictive values of severity models

The predictive values for in-hospital death of the severity scoring systems using binary cut-offs for the 722 patients of age 16 years and above are shown in Table 3. Four hundred and eighty-

**Table 2. Factors associated with in-hospital mortality.**

| Variable | Number of patients n (%) N = 822 | Mortality $n^1/n^2$ (%) with the variable | Mortality $n^1/n^2$ (%) without the variable | Unadjusted odds ratio | 95% CI | p-value | Adjusted odds ratio* | 95% CI | p-value |
|---|---|---|---|---|---|---|---|---|---|
| Any severely deranged vital sign | 525 (64) | 276/525 (53) | 92/297 (30) | 2.5 | 1.8–3.3 | <0.001 | 1.9 | 1.4–2.6 | <0.001 |
| Treatment with inotrope or vasopressor | 189 (23) | 122/189 (65) | 246/633 (39) | 2.9 | 2.0–4.0 | <0.001 | 2.3 | 1.6–3.4 | <0.001 |
| Received cardiopulmonary resuscitation | 161 (20) | 99/161 (62) | 269/661 (41) | 2.3 | 1.6–3.3 | <0.001 | 1.7 | 1.2–2.6 | 0.006 |
| Treatment with mechanical ventilation | 574 (70) | 283/574 (49) | 85/248 (34) | 1.9 | 1.4–2.5 | <0.001 | 1.5 | 1.1–2.1 | 0.024 |
| Positive HIV status | 32 (4) | 15/32 (47) | 353/790 (45) | 1.1 | 0.5–2.2 | 0.807 | 0.8 | 0.4–1.6 | 0.464 |
| Capillary refill time >3seconds | 50(6) | 23/50 (46) | 345/772 (45) | 1.1 | 0.6–1.9 | 0.857 | 0.8 | 0.4–1.5 | 0.467 |
| Sex (Male) | 414 (50) | 194/414(47) | 174/408 (43) | 0.8 | 0.6–1.1 | 0.225 | 0.9 | 0.7–1.2 | 0.421 |
| Had surgery during hospital admission | 515 (63) | 206/515 (40) | 162/307 (53) | 0.6 | 0.4–0.8 | <0.001 | 0.5 | 0.4–0.7 | <0.001 |
| Age group >50 | 163 (20) | 87/163 (53) | 281/659 (43) | 1.5 | 1.1–2.2 | 0.014 | 1.4 | 1.0–2.1 | 0.059 |
| Emergency Admission | 712 (87) | 320/712 (45) | 48/110 (44) | 1.1 | 0.7–1.6 | 0.797 | 0.9 | 0.6–1.4 | 0.703 |
| Severely deranged temperature | 218 (27) | 105/218 (48) | 263/604 (44) | 1.2 | 0.9–1.6 | 0.240 | 1.0 | 0.7–1.4 | 0.8609 |

* Covariates adjusted for were: HIV status, mechanical ventilation, capillary refill time, cardiopulmonary resuscitation, treatment with inotrope/vasopressor, had surgery during hospital admission, admission type, deranged temperature, sex and age

$n^1$ = number of patients that died, $n^2$ = number of patients

**Table 3. Predictive values of severity score models for patients over 16 years.**

| | Number with critical score (%) N = 722 | Mortality $n^1/n^2$ (%) with critical score | Mortality $n^1/n^2$ (%) without critical score | Odds Ratio | p-value | 95% C.I | Sensitivity % (95%C.I) | Specificity % (95% C.I) | PPV % (95% C.I) | NPV % (95% C.I) |
|---|---|---|---|---|---|---|---|---|---|---|
| Any severely deranged vital sign | 481 (67) | 256/481 (53) | 79/241 (33) | 2.3 | <0.001 | 1.7–3.2 | 76 (72–81) | 42 (37–47) | 53 (49–58) | 67 (61–73) |
| NEWS Score = >7 | 545 (76) | 280/545 (51) | 55/177 (31) | 2.3 | <0.001 | 1.6–3.4 | 84% (79–87) | 32% (27–36) | 51 (47–56) | 69 (62–76) |
| qSofa = >2 | 267 (34) | 147/267 (55) | 188/455 (42) | 1.7 | <0.001 | 1.3–2.4 | 44 (39–49) | 69 (64–74) | 55 (49–61) | 59 (54–63) |
| UVA Score > = 5 | 296 (41.). | 154/296 (52) | 181/426 (43) | 1.5 | 0.012 | 1.1–2.0 | 46 (41–52) | 63 (58–68) | 52 (46–58) | 58 (53–62) |
| TOTAL Score > = 2 | 517 (72) | 258/517 (50) | 77/205 (38) | 1.7 | 0.003 | 1.2–2.3 | 77 (72–81) | 33 (28–38) | 50 (46–54) | 62 (55–69) |
| TROPICSScore > = 8 | 58 (8) | 32/58 (55) | 303/664 (46) | 1.5 | 0.164 | 0.9–2.5 | 10 (7–13) | 93 (90–96) | 55 (42–68) | 54 (51–58) |
| MIME score > = 2 | 474 (66) | 234/474 (49) | 101/248 (41) | 1.4 | 0.027 | 1.0–1.9 | 70 (65–75) | 38 (33–43) | 49 (44–54) | 59 (53–65) |

NEWS-. National Early Warning Score; qSOFA—quick Sequential Organ Failure Assessment UVA—Universal Vital signs Assessment (UVA); TOTAL score (Tachypnoea, Oxygen saturation, Temperature, Alert and Loss of independence); TROPICS—Tropical Intensive Care Score; MIME—Malawi Intensive care Mortality risk Evaluation model

one (68%) had one or more severely deranged vital sign. Sensitivity of any severely deranged vital sign for in-hospital death was 76% (95% C.I 72–81) and specificity 42% (95% C.I 37–47). The performance of the severity models did not change substantially following stratification of patients into medical and surgical, whether data were retrospectively or prospectively collected, or when excluding patients with missing data. This excluded TROPICS whose performance improved when only patients with complete data were analysed. (Tables 4 & 5, S5, S6 and S8 Tables).

The performances of the severity scoring systems when alternative cut-offs were used can be seen in S7 Table. The AUROCs for the severity scoring systems using all possible scores

**Table 4. Predictive values of severity score models for medical patients over 16 years.**

| | Number with critical score (%) N = 170 | Mortality N1/n2 (%) with critical score | Mortality $n^1/n^2$ (%) without critical score | Odds Ratio | p-value | 95% C.I | Sensitivity % (95%C.I) | Specificity % (95% C.I) | PPV % (95% C.I) | NPV % (95% C.I) |
|---|---|---|---|---|---|---|---|---|---|---|
| Any severely deranged vital sign | 125 (74) | 72/125 (58) | 20/45 (44) | 1.7 | 0.131 | 0.9–3.4 | 78 (68–86) | 32 (22–44) | 58 (48–66) | 56 (40–70) |
| NEWS Score = >7 | 132 (78) | 73/132 (55) | 19/38 (50) | 1.2 | 0.567 | 0.6–2.5 | 79 (70–87) | 24 (15–35) | 55 (46–64) | 50 (33–67) |
| qSofa = >2 | 71 (42) | 42/71 (59) | 50/99 (51) | 1.4 | 0.265 | 0.8–2.6 | 46 (35–56) | 63 (51–73) | 59 (47–71) | 49.5 (39–60) |
| UVA Score > = 5 | 64 (38) | 35/64 (55) | 57/106 (54) | 1.03 | 0.908 | 0.6–1.9 | 38 (28–49) | 63 (51–74) | 55 (42–67) | 46 (37–56) |
| TOTAL Score > = 2 | 128 (75) | 72/128 (56) | 20/42 (48) | 1.4 | 0.331 | 0.7–2.8 | 78 (64–86) | 28 (19–40) | 56 (47–65) | 52 (36–68) |
| TROPICS Score > = 8 | 7 (4) | 4/7 (57) | 88/163 (54) | 1.1 | 0.870 | 0.2–5.2 | 4 (1–11) | 96 (89–99) | 57(18–90) | 46 (38–54) |
| MIME score > = 2 | 129 (76) | 73/129 (57) | 19/41 (46) | 1.5 | 0.253 | 0.7–3.1 | 79 (70–87) | 28 (19–40) | 57 (48–65) | 54 (37–69) |

**Table 5. Predictive values of severity score models for surgical patients over 16 years.**

| | Number with critical score (%) N = 552 | Mortality N1/n2 (%) with critical score | Mortality $n^1/n^2$ (%) without critical score | Odds Ratio | p-value | 95% C.I | Sensitivity % (95%C.I) | Specificity % (95% C.I) | PPV % (95% C.I) | NPV % (95% C.I) |
|---|---|---|---|---|---|---|---|---|---|---|
| Any severely deranged vital sign | 356 (64) | 184/356(52) | 59/196 (30) | 2.5 | 0.000 | 1.7–3.6 | 76 (70–81) | 44 (39–50) | 52 (46–57) | 70 (63–76) |
| NEWS Score = >7 | 413 (75) | 207/413 (50) | 36/139 (26) | 2.9 | 0.000 | 1.9–4.4 | 85 (80–89) | 33 (28–39) | 50(45–55) | 74 (66–81) |
| qSofa = >2 | 196 (36) | 105/196 (54) | 138/356 (39) | 1.8 | 0.001 | 1.3–2.6 | 43 (37–50) | 29(24–34) | 33 (27–38) | 40 (33–46) |
| UVA Score > = 5 | 232 (42) | 119/232 (51) | 124/320 (39) | 1.7 | 0.003 | 1.2–2.3 | 49 (43–55) | 63 (58–69) | 51 (45–58) | 61 (56–67) |
| TOTAL Score > = 2 | 389 (71) | 186/389 (48) | 57/163 (35) | 1.7 | 0.006 | 1.2–2.5 | 77 (71–82) | 34(29–40) | 48(43–53) | 65 (57–72) |
| TROPICS Score > = 8 | 51 (9) | 28/51 (55) | 215/501 (43) | 1.6 | 0.103 | 0.9–2.9 | 12 (8–16) | 93 (89–95) | 55(40–69) | 57(53–62) |
| MIME score > = 2 | 345 (63) | 161/345 (47) | 82/207 (40) | 1.3 | 0.106 | 0.9–1.9 | 66(60–72) | 41(35–46) | 47(41–52) | 60 (53–67) |

were 0.60 for any severely deranged vital sign, 0.59 for NEWS, 0.57 for qSOFA, 0.57 for UVA, 0.56 for TOTAL, 0.54 for TROPICS and 0.57 for MIME.

## Discussion

We have found that the presence of one or more severely deranged vital sign at admission to ICU in a tertiary hospital in Malawi was associated with in-hospital mortality. In addition, treatment with inotrope or vasopressor, having received cardiopulmonary resuscitation, treatment with mechanical ventilation, and not having had surgery before admission were associated with in-hospital mortality.

The study population had an ICU mortality of 35% and in-hospital mortality rate of 45%. These figures are high but are in-keeping with ICU mortality rates of 27%-64% reported elsewhere in Africa [14, 21–25]. ICU mortality in high resource settings have been reported to range from 8 to 17% [26]. A worldwide survey of ICUs in 2014 found an in-hospital mortality of 22% among the 10,069 included patients [26]. The high mortality rates in our unit could be due to either the admission of severely ill patients with poor prognosis or challenges with provision of good quality of care on the unit. As the majority (65%) of patients had one or more severely deranged vital sign at admission, this may support the first explanation. This proportion of patients is similar to the 69% of patients in an ICU in Tanzania (12). Furthermore, 10.6% of the patients in our study died within 24 hours of admission—many of these patients may not have been salvageable.

Other factors that were associated with mortality were treatment with an inotrope or vasopressor, mechanical ventilation and the use of cardiopulmonary resuscitation in the first hour after admission to the ICU. These factors could be seen as markers of disease severity and failing body physiology. In this cohort, 65% who received vasopressors on arrival to the ICU died. Vasopressors are commonly used in patients in shock. While the underlying cause of the shock leading to vasopressor treatment varied, international guidelines on management of shock include early identification, fluid resuscitation and early usage of vasopressors if fluid unresponsive [27–29]. A meta-analysis of randomized clinical trials (28,280 patients from 177 trials) looking at the effect of inotropes and vasopressors on mortality reported a mortality of

32% in those receiving inotropes/vasopressors [30]. The higher mortality in our study could be due to patients being admitted to ICU at a more advanced stage of their disease process.

Seventy percent of the patients in our unit received mechanical ventilation at admission. This is higher than comparisons with a hospital in Uganda where 18.7% of intensive care patients were mechanically ventilated [31] and in a survey of 361 intensive care units in 20 countries, where 33% of patients received mechanical ventilation [32]. In our unit the main indication for admission is the requirement/need for mechanical ventilation. Critically ill patients requiring mechanical ventilation have been previously found to have higher mortality than those not requiring mechanical ventilation [33, 34].

Globally, reported success rates of cardio-pulmonary resuscitation (CPR) for cardiac arrests in hospital and ICU vary widely, from initial success rates of 16–44% and long-term survival to discharge from hospital of 3–16% [35–39]. Outcome of CPR among in-hospital patients has been reported to be dependent on the early recognition and early initiation of basic life support, early defibrillation as well as prompt institution of advanced cardiac life support [10, 40]. Patients with reversible clinical conditions tend to have better outcome following cardiac arrest and CPR. A high proportion of our patients (20%) received cardiopulmonary resuscitation. Majority were surgical patients admitted from theatre This may reflect the illness severity of the patients on admission to the ICU as well as the quality anaesthetic and surgical services. Thirty-nine percent of the patients in this study that underwent cardiopulmonary resuscitation survived hospital admission. A possible reason for the high success rate may be the young population of patients (median age 31) and few comorbidities such as diabetes mellitus, hypertension or cardiac disease. In the United States, the mean age of patients with in-hospital cardiac arrest is 66 years [39]. Alternatively, a large proportion of the cardiac arrests may have been due to reversible causes, such as a lack of timely identification of an acute deterioration, equipment failure, challenges transporting ill patients from admitting ward to the ICU or other potentially avoidable reasons.

Critical illness as defined by binary cut-offs in each of the severity models had an association with in-hospital mortality (statistically significant in all models except TROPICS). However, in our opinion the performances of the models were too low to be clinically useful as sole factors for individual patient decisions. The performance of models may be possible to increase if all possible values of the model were used, rather than binary scores, but even then the AUROCs in this study were low—between 0.54 and 0.60. The trade-off between performance and usefulness is a challenging one that we have discussed previously [41]. In this study, an a-priori decision based on perceived clinical feasibility and usefulness was made to convert compound scores into binary scores. Additionally, the patients in this study were heterogeneous and the majority were admitted from the operating theatres and emergency departments, with different characteristics than the populations used to derive the scores. Those from theatre may have residual effects of the anaesthesia affecting their vital signs, and patients may have undergone resuscitation and stabilisation of deranged vital signs before transfer to ICU. Other studies have found different performance of the models: in another population in Malawi, qSOFA was found to have a sensitivity of 88% and a specificity of 30.3% [42]. In South Asia, TROPICS had a sensitivity of 70% and a specificity of 69% [18]. One limitation of the TROPICS model in this study is that it requires laboratory values hemoglobin and urea which can be challenging to obtain in our unit. We had very limited data on hemoglobin and urea, and this might have affected performance. The models in this study can be useful for assessing the illness severity of patient cohorts for benchmarking and other comparisons. The key would be ensuring good quality data collection. The missing data in this study may have impacted the performance of the models. Though some of these models were developed utilizing non-ICU patient populations such as emergency departments and medical wards, they are

still useful to identify severely ill patients. The component variables for the models can be easily collected during the clinical care of an ICU patient in a low-resource setting and do not require invasive monitors or laboratory measurements, as would APACHE, SAPS or SOFA [7, 43].

The simplest system—use of one or more severely deranged vital sign—performed as well as any of the compound scoring systems in our heterogeneous ICU population. While one study found that compound scores are better than single parameters for predicting poor outcome [44, 45], other studies have found conflicting results [14, 20, 22]. Compound scores have the negative attributes of requiring additional work for the health staff to add up the scores and a risk of incorrect addition [46]. Another challenge with compound scores is that they are difficult to use in settings where all the components are not used in routine clinical care and documentation quality is poor. Additionally, compound scores are unable to indicate potential appropriate treatments, in contrast to single severely deranged vital signs such as suggesting optimising oxygen therapy when hypoxia is identified [47].

Our study has several strengths. We included all patients admitted to the ICU during the study period, providing a large database of patients in a setting from which data are sparse. Our prospective data collection reduced the amount of missing data. The study's limitations include that it was single-centered study from one teaching hospital and as such findings should be transferred to other settings with caution. We were unable to include data from the ongoing care of the patients in the ICU. Some of the data were collected retrospectively and there were some missing data, especially for GCS (42% of patients were lacking a GCS score) and for capillary refill time. The sensitivity analyses conducted showed no marked variations on estimates when missing data were handled in different ways, apart from for capillary refill time. Transferring the findings from this study to other patient populations should be done with caution given that they were a heterogeneous ICU cohort.

## Conclusion

A mixed cohort of patients admitted to an ICU in Malawi with one or more severely deranged vital sign had a high risk of dying—this simple indicator could be a marker of an increased risk of death that could be useful for clinical decision making together with other clinical information. Several other simple parameters and treatments were also found to be associated with an increased risk of death.

## Supporting information

**S1 Table. Sensitivity analyses of the factors associated with in-hospital mortality when missing data are handled by i) only including patients with complete data and ii) imputed as deranged.**
(DOCX)

**S2 Table. Description of models used in the study.**
(DOCX)

**S3 Table. Univariable logistic regression of individual vital signs.**
(DOCX)

**S4 Table. Percentage of data missing per variable.**
(DOCX)

**S5 Table. Predictive values of severity score models for patients over 16 years included in the prospective data collection period.**
(DOCX)

**S6 Table. Predictive values of severity score models for patients over 16 years included in the retrospective data collection period.**
(DOCX)

**S7 Table. Predictive values of the scoring models for patients over 16 years with different cut offs for defining critical illness used.**
(DOCX)

**S8 Table. Predictive values of severity score models for patients over 16 years using cases with complete data only.**
(DOCX)

## Acknowledgments

Special thanks to all nurses in the Queen Elizabeth Central Hospital adult ICU for their role in data collection and support.

## Author Contributions

**Conceptualization:** Mtisunge Kachingwe, Raphael Kazidule Kayambankadzanja, Singatiya Stella Chikumbanje, Tim Baker.

**Data curation:** Mtisunge Kachingwe, Raphael Kazidule Kayambankadzanja.

**Formal analysis:** Mtisunge Kachingwe, Raphael Kazidule Kayambankadzanja.

**Methodology:** Mtisunge Kachingwe.

**Project administration:** Raphael Kazidule Kayambankadzanja, Tim Baker.

**Supervision:** Raphael Kazidule Kayambankadzanja, Wezzie Kumwenda Mwafulirwa, Tim Baker.

**Writing – original draft:** Mtisunge Kachingwe.

**Writing – review & editing:** Mtisunge Kachingwe, Raphael Kazidule Kayambankadzanja, Wezzie Kumwenda Mwafulirwa, Singatiya Stella Chikumbanje, Tim Baker.

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
