## [Decision Letter · Decision Letter 0]

8 Nov 2021

PONE-D-21-17302Factors associated with poor outcomes in patients in an Intensive Care Unit in a tertiary hospital in MalawiPLOS ONE

Dear Dr. Kachingwe,

Thank you for submitting your manuscript to PLOS ONE. After careful consideration, we feel that it has merit but does not fully meet PLOS ONE’s publication criteria as it currently stands. Therefore, we invite you to submit a revised version of the manuscript that addresses the points raised during the review process. We appreciate the importance of your work. Please find the reviewers comments below that largely identify questions in your methodology requiring clarification.

We look forward to your resubmission.

We look forward to receiving your revised manuscript.

Kind regards,

Regan Marsh, MD, MPH

Academic Editor

PLOS ONE

Journal Requirements:

2. Thank you for providing the date(s) when patient medical information was initially recorded. Please also include the date(s) on which your research team accessed the databases/records to obtain the retrospective data used in your study.

5. Please amend the manuscript submission data (via Edit Submission) to include author Singatiya Stella Chikumbanje.

7. Please upload a copy of Figure 2, to which you refer in your text on page 22. If the figure is no longer to be included as part of the submission please remove all reference to it within the text.

Reviewers' comments:

Reviewer's Responses to Questions

**Comments to the Author**

1. Is the manuscript technically sound, and do the data support the conclusions?

Reviewer #1: Partly

Reviewer #2: Partly

2. Has the statistical analysis been performed appropriately and rigorously? 

Reviewer #1: No

Reviewer #2: I Don't Know

3. Have the authors made all data underlying the findings in their manuscript fully available?

Reviewer #1: No

Reviewer #2: No

4. Is the manuscript presented in an intelligible fashion and written in standard English?

Reviewer #1: Yes

Reviewer #2: Yes

5. Review Comments to the Author

Reviewer #1: Thank you for asking me to review the manuscript, “Factors associated with poor outcomes in patients in an Intensive Care Unit in a tertiary hospital in Malawi.” The study covers an understudied and important topic of mortality prediction in critically ill patients from sub-Saharan Africa. The manuscript is well written and a nice contribution to the field. However, the authors may wish to consider the following:

Title

What is meant by “poor outcomes?” Since the study was designed to identify predictors of death in the ICU, would it not be better to explicitly state this in the title?

Methodology

What was the reason for the study design that included both retrospective and prospective data collection? This study design likely had an impact on the amount of missing data, which likely influenced the results of the analysis.

The authors state that data were collected from within one hour of admission to the ICU. Can they provide data regarding time from admission as this is time frame used to derive most of the severity scores tested.

Please provide more details about what imputation strategy was used and why the strategy was chosen.

A sensitivity analysis is described based on analysis of patients with complete data; however, I cannot find the results of the sensitivity analysis in the Results section of the manuscript. Was the sensitivity analysis done?

The greatest limitation of the study is found in the secondary analysis of severity scores. In this analysis, only sens/spec/ppv/npv was assessed using binary cutoffs from the scores. However, this analysis misses an opportunity to better assess the scores for their ability to discriminate for the outcome of death via the calculation of area under the receive operating characteristic curve. AUCs provide a better assessment of the overall discrimination ability of the tests. Using binary cutoffs reduces statistical power and provides a limited assessment particularly when the cutoffs used are essentially arbitrary and not validated in the population being analyzed, as in this case.

Furthermore, the argument that clinicians in LMICs are not capable of calculating simple scores does not carry much water, particularly in ICUs and when smart phones and smart phone apps, which could calculate scores, are so ubiquitous in LMICs including those in Africa. Cutoffs can be useful, but individual clinicians need to assess and validate their priorities for maximizing sens/spec before assessing specific binary cut-offs for clinical use. As well, ppv/npv are dependent upon prevalence of the outcome of the population. Accordingly, if binary cut-offs are to be included, they should not be limited to arbitrary cutoffs, but instead a table should be provided with columns for sens/spec/ppv/npv and rows for each cseverity risk score starting from 0 to the maximum calculated score. As well, AUCs should be calculated for each severity risk score and for the assessment of the aggregate of single vital sign abnormalities.

An additional significant limitation of this study is the application of severity risk scores to a heterogeneous clinical population that is different from the original derivation cohorts of the severity risk scores. In this study, the population was almost 50% post-surgical and clinically heterogeneous, which does not reflect the derivation population cohorts of the severity risk scores. Also, the scores were calculated at the time of admission to the ICU, not necessarily at the time of admission to hospital. Can the authors compare the performance of scores calculated at admission to hospital to those calculated at the time of admission to the ICU? Can the authors stratify the analyses based on medical vs surgical patients and whether data were retrospectively or prospectively collected? Also many patients were transferred from other hospitals so their physiology may already have been altered by preceding resuscitation which would impact the performance of severity risk scores. All of these issues should be detailed as limitations of the study in the Discussion

Results

Please provide a table of % missingness for each clinical variable/predictor used in the analyses. Please also provide data regarding how many patients were excluded from the analysis due to missing outcome data.

Please provide data for the pre-planned sensitivity analysis of patients with complete data

As above, please provide results of AUCs for severity risk scores and aggregate abnormal vital signs

As above, please provide results for sens/spec/ppv/npv for each calculated score of the different severity scores starting at ‘0.’

For the assessment of individual risk factors for death, why was age categorized as < or >50 years? This seems to be an arbitrary cutoff which will lead to decreased statistical power. Age should probably be analyzed as a continuous variable.

Consider creating a figure with a line graph of %mortality plotted against severity risk score results and a similar figure with OR 95%CI plotted against risk score results.

Reviewer #2: Using a combination of retrospective and prospective data, Mtisunge Kachingwe and colleagues aim to identify predictors of poor outcomes in patients admitted to an ICU in Malawi.

My primary comments are related to the data collection and definition of variables and the secondary aim (comparing predictive value of severity scoring systems within this cohort).

Major comments

Methodology

1. Additional details on data extraction would be helpful. What prospective data were extracted using the data sheet? What variables were abstracted from retrospective chart review? How was time of ICU admission determined? If multiple sets of vital signs were recorded within the first hour of ICU admission, which set of vital signs was included in analyses? How were laboratory values extracted? Were laboratory studies only included if they were drawn within the first hour of admission?

2. The authors state that data were extracted from the departmental electronic database of all ICU patients” and “the data were collected prospectively from December 2017 by the nurses at admission or within 1 hour of admission, using a paper-based data collection tool.”

a. In the Data Management section, it is not clear if the variables of interest are restricted to the first hour of admission. For example, for a patient to be considered to have received mechanical ventilation in data analyses, did they have to be ventilated during the first hour of admission or did they count ventilation at any point during ICU admission?

b. Was retrospective data collection restricted to the first hour of admission?

3. Lines 128 to 137: The section on comparison scoring systems (e.g. UVA, NEWS, TropICS) would benefit from additional clarification and explanation.

a. The primary outcome of this study was in-hospital mortality. However, many of the comparison scoring systems were derived and validated to predict different outcomes (e.g. 24 hour mortality, ICU mortality, 3 day mortality) among different populations (e.g. patients on medical wards) from the study population. This should be acknowledged and justified.

b. The authors state that “models were chosen for their greater potential feasibility in low resourced settings.” If this is the case, why did the authors select the NEWS instead of the MEWS which has been studied in Uganda? (PLoS ONE 11(3):e0151408. doi:10.1371/journal.pone.0151408)

c. The authors cite the Modified-MPM from Rwanda later in the manuscript, but do not include it as a reference scoring system. What was the reason for this?

Results

1. No information is provided about the number of records with missing data which were imputed as normal. Also no information provided about the results of the sensitivity analysis discussed in line 125.

2. The relatively low odds ratio for CPR is very interesting. Additionally, the proportion of patients receiving CPR within the first hour of ICU admission is quite high (20%). Additional information about how receipt of CPR was defined for the purposes of these analyses as well as additional data about the patients who received CPR would help interpret this finding.

Discussion

1. Line 261: I do not think the analyses support broad conclusions about “the predictive performance” of the models. Many of the models were evaluated for their ability to predict outcomes they were not designed to predict. This conclusion should be more qualified.

2. The limitation that some data were collected in a retrospective manner should be addressed.

Minor comments

Methodology

1. Line 72: The authors refer to the “main” ICU as the site of data collection. Is information available on the other ICUs in the facility and the types of patients they admit? This would help to understand any potential biases associated with limiting inclusion criteria to one of multiple ICUs at the facility.

2. Line 133: The authors include a citation for a commentary on the NEWS but not for the primary study and the TOTAL score manuscript should be cited here but is not cited until later in the manuscript.

3. Line 143: Authors state that “each patient’s qSOFA, UVA, NEWS, MIME and TROPICS scores were calculated from their vital signs.” However, UVA and TropICS include non-vital sign data.

Results

1. Line 157: “Other hospitals” is listed as a source of admissions to the ICU but this category is not included in Table 1.

2. Given the patient population is very heterogenous the authors may want to consider reporting odds ratios by the subgroups of surgical and non-surgical patients.

6. PLOS authors have the option to publish the peer review history of their article (what does this mean?). If published, this will include your full peer review and any attached files.

Reviewer #1: No

Reviewer #2: No

---

## [Author Response · Author response to Decision Letter 0]

11 Jan 2022

Reviewer #1: 

Thank you for asking me to review the manuscript, “Factors associated with poor outcomes in patients in an Intensive Care Unit in a tertiary hospital in Malawi.” The study covers an understudied and important topic of mortality prediction in critically ill patients from sub-Saharan Africa. The manuscript is well written and a nice contribution to the field. However, the authors may wish to consider the following.

We thank the reviewer and have modified the manuscript to address the points made. We believe that the manuscript is now more readable, more informative, and its conclusions more useful to the public

1. What is meant by “poor outcomes?” Since the study was designed to identify predictors of death in the ICU, would it not be better to explicitly state this in the title?

Thank you. The title has been amended to – Factors associated with in-hospital mortality of patients admitted to an Intensive Care Unit in a tertiary hospital in Malawi 

Methodology

2. What was the reason for the study design that included both retrospective and prospective data collection? This study design likely had an impact on the amount of missing data, which likely influenced the results of the analysis.

Thank you for your comment. We have included all the patients from the departmental database, which includes those entered retrospectively from January to November 2017. We believe the study is stronger for the inclusion of these additional patients. We have added text in the material and methods to clarify this.

3. The authors state that data were collected from within one hour of admission to the ICU. Can they provide data regarding time from admission as this is time frame used to derive most of the severity scores tested?

The vital signs documented were the first vital signs recorded on admission to the ICU, and the severity scores were based on these vital signs, not on admission to hospital. We do not have data on time of admission to hospital. Please let us know if this answers your query, or if we have misunderstood.

4. Please provide more details about what imputation strategy was used and why the strategy was chosen.

We have provided more details in the data analysis section in the material and methods. 

5. A sensitivity analysis is described based on analysis of patients with complete data; however, I cannot find the results of the sensitivity analysis in the Results section of the manuscript. Was the sensitivity analysis done?

Apologies. We have now included the sensitivity analyses in supplementary table 1

6. The greatest limitation of the study is found in the secondary analysis of severity scores. In this analysis, only sens/spec/ppv/npv was assessed using binary cutoffs from the scores. However, this analysis misses an opportunity to better assess the scores for their ability to discriminate for the outcome of death via the calculation of area under the receive operating characteristic curve. AUCs provide a better assessment of the overall discrimination ability of the tests. Using binary cutoffs reduces statistical power and provides a limited assessment particularly when the cutoffs used are essentially arbitrary and not validated in the population being analyzed, as in this case. Furthermore, the argument that clinicians in LMICs are not capable of calculating simple scores does not carry much water, particularly in ICUs and when smart phones and smart phone apps, which could calculate scores, are so ubiquitous in LMICs including those in Africa. Cutoffs can be useful, but individual clinicians need to assess and validate their priorities for maximizing sens/spec before assessing specific binary cut-offs for clinical use. As well, ppv/npv are dependent upon prevalence of the outcome of the population. Accordingly, if binary cut-offs are to be included, they should not be limited to arbitrary cutoffs, but instead a table should be provided with columns for sens/spec/ppv/npv and rows for each severity risk score starting from 0 to the maximum calculated score. As well, AUCs should be calculated for each severity risk score and for the assessment of the aggregate of single vital sign abnormalities.

We agree that this is a really important issue and AUROCs can provide additional information. Indeed, in previous work we have studied AUROCs (for example Kayambankadzanja RK et al. The Prevalence and Outcomes of Sepsis in Adult Patients in Two Hospitals in Malawi. The American journal of tropical medicine and hygiene 2020, Baker T, et al. Single Deranged Physiologic Parameters Are Associated with Mortality in a Low-Income Country. Critical care medicine 2015; 43(10): 2171-9.)

However, for this study, we made the a-priori decision to use binary cut-offs for the scoring systems. This is due to our clinical and systems-based experience that in low-resourced environments with few staff, calculating and using compound scores is not feasible. In the settings where we have worked, including in the hospital in this study, when we have attempted to introduce compound scoring systems we have seen calculation errors, misclassifications, and after some time the scores are not used at all. Additionally, compound scores are unable to indicate potential appropriate treatments, in contrast to single severely deranged vital signs such as suggesting optimizing oxygen therapy when hypoxia is identified. Our aims in this study were to identify factors associated with in-hospital death and to study a binary score (single severely deranged vital signs) as we hypothesized that it could be useful and feasible to use, and compare it to other scores that have been converted to binary scores. The comparison of scoring systems was not the main aim of the study, and we are reluctant to add additional tables and performance measures as it could distract away from the main aim of the study. 

We have added text about this in the discussion. We hope this is reasonable

7. An additional significant limitation of this study is the application of severity risk scores to a heterogeneous clinical population that is different from the original derivation cohorts of the severity risk scores. In this study, the population was almost 50% post-surgical and clinically heterogeneous, which does not reflect the derivation population cohorts of the severity risk scores. Also, the scores were calculated at the time of admission to the ICU, not necessarily at the time of admission to hospital. Can the authors compare the performance of scores calculated at admission to hospital to those calculated at the time of admission to the ICU? Can the authors stratify the analyses based on medical vs surgical patients and whether data were retrospectively or prospectively collected? Also many patients were transferred from other hospitals so their physiology may already have been altered by preceding resuscitation which would impact the performance of severity risk scores. All of these issues should be detailed as limitations of the study in the Discussion

Thank you for these good comments. We have added information to the discussion about these points. 

We do not have data on the patients’ conditions on admission to hospital. 

We have added tables 4 and 5 which show results of the models after stratification based on medical vs surgical patients 

We have also added supplementary tables 5 & 6 showing the results after stratification based on prospective and retrospective data

The methods and results sections have been updated to report on these further analyses

Results

8. Please provide a table of % missingness for each clinical variable/predictor used in the analyses. Please also provide data regarding how many patients were excluded from the analysis due to missing outcome data.

Table with percentage of missing data has been added as a supplementary table 4

Two patients were excluded from analysis due to missing outcome data (start of results section)

9. Please provide data for the pre-planned sensitivity analysis of patients with complete data

Thank you. This is in supplementary table 1 

10. As above, please provide results of AUCs for severity risk scores and aggregate abnormal vital signs

Thank you for your comment. We have responded in the above comment about AUROCs. We hope this is reasonable and acceptable.

11. As above, please provide results for sens/spec/ppv/npv for each calculated score of the different severity scores starting at ‘0.’

Thank you. We have responded to this in the above comment.

12. For the assessment of individual risk factors for death, why was age categorized as < or >50 years? This seems to be an arbitrary cutoff which will lead to decreased statistical power. Age should probably be analyzed as a continuous variable.

Thank you for the comment. A decision to convert age to binary was made a-priori for feasibility reasons and in keeping with the rest of the factors, even though we agree some statistical power is lost. Malawi has a young population hence the a-priori decision to categorize with a cut of 50.

13. Consider creating a figure with a line graph of %mortality plotted against severity risk score results and a similar figure with OR 95%CI plotted against risk score results.

Thank you for this recommendation. For this study we did not analyze for severity risk scores. We hope we understood the comment and that is acceptable.

Reviewer #2: 

My primary comments are related to the data collection and definition of variables and the secondary aim (comparing predictive value of severity scoring systems within this cohort).

We thank the reviewer and have modified the manuscript to address the points made. We believe that the manuscript is now more readable, more informative, and its conclusions more useful to the public

Methodology

1. Additional details on data extraction would be helpful. What prospective data were extracted using the data sheet? What variables were abstracted from retrospective chart review? How was time of ICU admission determined? If multiple sets of vital signs were recorded within the first hour of ICU admission, which set of vital signs was included in analyses? How were laboratory values extracted? Were laboratory studies only included if they were drawn within the first hour of admission?

Thank you for your comments. We have added additional details to the materials and method section, data collection section. All patients admitted to the ICU during the study period were included as study participants. In December 2017, the department created an electronic database of all admissions, including data entered from the paper records for patients from 1st January 2017 and prospectively from December 2017. The prospective data were collected by the nurses immediately at admission or within 1 hour of admission if there were logistical delays in documentation, using a paper-based data collection tool. The data collection tool collected data on demographics, diagnosis, and ward from which they were admitted, vital signs, lab investigations and interventions received. The data extracted from the patients admitted between January and November included the first vital signs recorded during the first hour of admission to the ICU. Lab investigation results were obtained from the file. Data collection was supervised for quality in the ICU and double-data entered into the database. Follow-up of patients continued on the wards until hospital discharge or death. The primary endpoint for the study was in-hospital death. Patients lacking data on hospital outcome were excluded from analysis.

2. The authors state that data were extracted from the departmental electronic database of all ICU patients” and “the data were collected prospectively from December 2017 by the nurses at admission or within 1 hour of admission, using a paper-based data collection tool.”

a. In the Data Management section, it is not clear if the variables of interest are restricted to the first hour of admission. For example, for a patient to be considered to have received mechanical ventilation in data analyses, did they have to be ventilated during the first hour of admission or did they count ventilation at any point during ICU admission?

b. Was retrospective data collection restricted to the first hour of admission?

Thank you. This has been clarified in the data management section. The variable data was obtained from the data collection tool in which information was recorded within an hour of admission.

We have now clarified this in the data collection section. The retrospective data was also restricted to admission or within 1 hour of admission

3. Lines 128 to 137: The section on comparison scoring systems (e.g. UVA, NEWS, TropICS) would benefit from additional clarification and explanation.

A description of the models is now provided in supplementary table 2. 

a. The primary outcome of this study was in-hospital mortality. However, many of the comparison scoring systems were derived and validated to predict different outcomes (e.g. 24 hour mortality, ICU mortality, 3 day mortality) among different populations (e.g. patients on medical wards) from the study population. This should be acknowledged and justified.

Thank you for this comment. We agree with the reviewers on this observation and have acknowledged this in the discussion. Assessments of the performances of the severity scoring systems has limitations as the population in this study had different case-mix characteristics and was exclusively an ICU population

b. The authors state that “models were chosen for their greater potential feasibility in low resourced settings.” If this is the case, why did the authors select the NEWS instead of the MEWS which has been studied in Uganda? (PLoS ONE 11(3):e0151408. doi:10.1371/journal.pone.0151408)

Thank you for your comment. We opted for NEWS as it more widely used. There are some studies that suggest that NEWS is better for example; Smith et al The ability of the National Early Warning Score (NEWS) to discriminate patients at risk of early cardiac arrest, unanticipated intensive care unit admission, and death, Resuscitation 84 (2013) 465–470

c. The authors cite the Modified-MPM from Rwanda later in the manuscript, but do not include it as a reference scoring system. What was the reason for this?

Thank you. We considered use of MPM score from Rwanda however we opted not to use it as we anticipated challenges to collect data on the variable of “confirmed or suspected infection”. 

Results

4. No information is provided about the number of records with missing data which were imputed as normal. Also no information provided about the results of the sensitivity analysis discussed in line 125.

Table with amount of missing variable has been added as supplementary table 4

5. The relatively low odds ratio for CPR is very interesting. Additionally, the proportion of patients receiving CPR within the first hour of ICU admission is quite high (20%). Additional information about how receipt of CPR was defined for the purposes of these analyses as well as additional data about the patients who received CPR would help interpret this finding.

Thank you. We have amended the materials and method (data analysis) section to define CPR. Cardiopulmonary resuscitation consisted of advanced life support (ALS) to patients who suffered a cardiac arrest. (10) Cardiac arrest was defined in the Utstein style as “the cessation of cardiac mechanical activity confirmed by the absence of a detectable pulse, unresponsiveness, and apnea (or agonal respirations)”

The discussion section has been amended as follows

A high proportion of our patients (20%) received cardiopulmonary resuscitation. Majority were surgical patients admitted from theatre. This may reflect the illness severity of the patients on admission to the ICU as well as the quality anesthetic and surgical services. Thirty-nine percent of the patients in this study that underwent cardiopulmonary resuscitation survived hospital admission. A possible reason for the high success rate may be the young population of patients (median age 31) and few comorbidities such as diabetes mellitus, hypertension or cardiac disease. In the United States, the mean age of patients with in-hospital cardiac arrest is 66 years (39). Alternatively, a large proportion of the cardiac arrests may have been due to reversible causes, such as a lack of timely identification of an acute deterioration, equipment failure, challenges transporting ill patients from admitting ward to the ICU or other potentially avoidable reasons

Discussion

6. Line 261: I do not think the analyses support broad conclusions about “the predictive performance” of the models. Many of the models were evaluated for their ability to predict outcomes they were not designed to predict. This conclusion should be more qualified.

Thank you. The text and the conclusion have been modified.

7. The limitation that some data were collected in a retrospective manner should be addressed.

Thank you. This has been added to the discussion section.

Methodology

8. Line 72: The authors refer to the “main” ICU as the site of data collection. Is information available on the other ICUs in the facility and the types of patients they admit? This would help to understand any potential biases associated with limiting inclusion criteria to one of multiple ICUs at the facility.

The hospital has two ICUs. The second one recently established caters for pediatric patients and was not included in this study. Occasionally the adult ICU will admit pediatric patients when the pediatric ICU is full. We have changed from “main” to “adult” in the manuscript and clarified in the manuscript. 

9. Line 133: The authors include a citation for a commentary on the NEWS but not for the primary study and the TOTAL score manuscript should be cited here but is not cited until later in the manuscript.

Thank you for this observation. The citing of TOTAL has been amended.

For NEWS the following article is cited as Royal College of Physicians. National Early Warning Score ( NEWS ) - Standardising the assessment of acute-illness severity in the NHS. Report of a working party. 2012. 47

We hope we understood this comment correctly.

10. Line 143: Authors state that “each patient’s qSOFA, UVA, NEWS, MIME and TROPICS scores were calculated from their vital signs.” However, UVA and TropICS include non-vital sign data.

Amended the terminology to variables

Results

11. Line 157: “Other hospitals” is listed as a source of admissions to the ICU but this category is not included in Table 1.

Table has been amended to say other hospitals 

12. Given the patient population is very heterogenous the authors may want to consider reporting odds ratios by the subgroups of surgical and non-surgical patients.

Thank you. We have added tables 4 and 5 which report results based on surgical and medical patients.

---

## [Decision Letter · Decision Letter 1]

12 May 2022

PONE-D-21-17302R1

Factors associated with in-hospital mortality of patients admitted to an Intensive Care Unit in a tertiary hospital in Malawi

PLOS ONE

Dear Dr. Mtisunge Kachingwe,

Thank you for submitting your manuscript to PLOS ONE. After careful consideration, we feel that it has merit but does not fully meet PLOS ONE’s publication criteria as it currently stands. Therefore, we invite you to submit a revised version of the manuscript that addresses the points raised during the review process.

We appreciate your efforts for the study and the authors have made a careful revision to the manuscript. However, there are some important points that are required to define clearly. Please carefully respond to the reviewers’ comments and suggestions particularly the AUC for each score and some points in the discussion.

We look forward to receiving your revised manuscript.

Kind regards,

Vipa Thanachartwet, M.D.

Academic Editor

PLOS ONE

Reviewers' comments:

Reviewer's Responses to Questions

**Comments to the Author**

1. If the authors have adequately addressed your comments raised in a previous round of review and you feel that this manuscript is now acceptable for publication, you may indicate that here to bypass the “Comments to the Author” section, enter your conflict of interest statement in the “Confidential to Editor” section, and submit your "Accept" recommendation.

Reviewer #1: (No Response)

Reviewer #2: (No Response)

2. Is the manuscript technically sound, and do the data support the conclusions?

Reviewer #1: Partly

Reviewer #2: Partly

3. Has the statistical analysis been performed appropriately and rigorously? 

Reviewer #1: Yes

Reviewer #2: Yes

4. Have the authors made all data underlying the findings in their manuscript fully available?

Reviewer #1: No

Reviewer #2: No

5. Is the manuscript presented in an intelligible fashion and written in standard English?

Reviewer #1: Yes

Reviewer #2: Yes

6. Review Comments to the Author

Reviewer #1: The authors have made a careful evaluation of the reviewer comments and have responded with appropriate revisions to the manuscript. However, concern remains about the use of cut-offs for the severity risk scores. Given the use of scores in a heterogeneous and unvalidated ICU patient population, the authors should provide more information about the performance of the scores. Sens/spec/ppv/npv can go up or down depending on the cut-off used and depending on the priority of the clinician. For example, in the data provided, NEWS >/=7 has a high sens and low spec; whereas, qSOFA >/=2 has low sens and high spec. Presumably, these could be reversed by simply adjusting the cut-offs for each score. Since the cut-offs are arbitrary and we don't know the optimal cut-off for each score evaluated in this population, the authors should provide sens/spec/ppv/npv for each value within each severity risk score so they can be appropriately evaluated. To use the prior example again, perhaps 7 and 2 are simply not the correct cut-offs (if one exists) in this population for NEWS and qSOFA, respectively. The overall AUC for each score would also be useful information and should be reported. If the authors are unable or unwilling to provide these data, then they should consider not including the secondary aim of evaluating severity risk scores. Finally, the large amount of missing GCS data should be further emphasized as a limitation particularly for GCS evaluation alone and for scores which include GCS, i.e. UVA. Accordingly, each risk score should be added to Supplementary Table 1 to show the effect of missing data on the performance of the risk scores. Ideally, this would include how AUC changes.

Reviewer #2: Dr. Kachingwe and colleagues have carefully revised their manuscript.

Major comments

Discussion

1) The second part of the Discussion is dedicated to deranged vital signs and severity scores as predictors of in-hospital mortality. While I agree with the authors’ central argument—that the presence of one or more severely deranged vital signs on admission to an ICU in Malawi is a feasible prognostic marker for increased risk of in-hospital death—I think this conclusion needs further qualification due to the heterogeneity of the study population and the rate of missing data. Lines 331 to 339 provide an excellent summary as well as clear and helpful context. I think this is the core of the part of the discussion and would consider streamlining the rest of this section.

a. Study population heterogeneity: The authors state that “assessments of the performances of the severity scoring systems has limitations as the population in this study had different case-mix characteristics.” However, this also seems to be applicable to single deranged vital signs. I would consider acknowledging this and including discussion on how this limitation may have influenced the results.

b. Missing data: Please expand on the limitation that “there are some missing data”. There were significant rates of missing data, for the variables capillary refill time and Glasgow coma scale in particular. By providing the two sensitivity analyses in supplementary table 1 (i.e., case wise deletion of missing data and imputed abnormal data), the authors do an excellent job helping the reader to understand the potential impact of missing data on their estimates. I think the manuscript would benefit from a more detailed discussion in the text comparing estimates from the primary analysis with the two sensitivity analyses in table S1.

1) The NEWS score and any severely deranged vital sign had very similar point estimates in the primary analysis (Table 3). It therefore strikes me as inconsistent to say that “performances of the models were too low to be clinically useful by themselves for individual patient decisions” and to also suggest that severely deranged vital signs have clinical utility.

Minor comments

1. The authors state: “critical illness as defined by the binary cut-offs in each of the severity models showed a significant association with in-hospital mortality.” However, not all of the severity model binary cut-offs had a significant association with in-hospital mortality.

2. Tables 4 and 5: Percentages are missing from the first data column

3. Supplementary table 1: What is the total number of patients with complete data included in these analyses?

4. The manuscript needs additional copy editing.

7. PLOS authors have the option to publish the peer review history of their article (what does this mean?). If published, this will include your full peer review and any attached files.

Reviewer #1: No

Reviewer #2: No

---

## [Author Response · Author response to Decision Letter 1]

24 Jul 2022

Dear Editorial board members, 

RE: Response to reviewers 

We thank the two reviewers for their comments on our manuscript titled “Factors associated with in-hospital mortality of patients admitted to an Intensive Care Unit in a tertiary hospital in Malawi”. Below is our response to each point raised by the reviewers. We hope that we satisfyingly addressed them and that the manuscript will be now suited for publication. 

Sincerely, 

On behalf of all authors, 

Dr Mtisunge Kachingwe

Reviewer #1: 

The authors have made a careful evaluation of the reviewer comments and have responded with appropriate revisions to the manuscript. However, concern remains about the use of cut-offs for the severity risk scores. Given the use of scores in a heterogeneous and unvalidated ICU patient population, the authors should provide more information about the performance of the scores. Sens/spec/ppv/npv can go up or down depending on the cut-off used and depending on the priority of the clinician. For example, in the data provided, NEWS >/=7 has a high sens and low spec; whereas, qSOFA >/=2 has low sens and high spec. Presumably, these could be reversed by simply adjusting the cut-offs for each score. Since the cut-offs are arbitrary and we don't know the optimal cut-off for each score evaluated in this population, the authors should provide sens/spec/ppv/npv for each value within each severity risk score so they can be appropriately evaluated. 

Thank you for this comment! We have further analysed the scoring systems using every possible cut-off for a critical score and present all the performance measures in Supplementary Table 7. 

To use the prior example again, perhaps 7 and 2 are simply not the correct cut-offs (if one exists) in this population for NEWS and qSOFA, respectively. The overall AUC for each score would also be useful information and should be reported. If the authors are unable or unwilling to provide these data, then they should consider not including the secondary aim of evaluating severity risk scores. 

Thank you for this. The AUCs for each scoring system has now been added to the results section.

Finally, the large amount of missing GCS data should be further emphasized as a limitation particularly for GCS evaluation alone and for scores which include GCS, i.e. UVA. Accordingly, each risk score should be added to Supplementary Table 1 to show the effect of missing data on the performance of the risk scores. Ideally, this would include how AUC changes.

We thank you for your comment. We have added text to the discussion to highlight the missing data, especially for GCS and capillary refill time. We have added a Supplementary Table 8 to show the performance of the severity scores with just complete data. 

Reviewer #2: Dr. Kachingwe and colleagues have carefully revised their manuscript.

Major comments

Discussion

1) The second part of the Discussion is dedicated to deranged vital signs and severity scores as predictors of in-hospital mortality. While I agree with the authors’ central argument—that the presence of one or more severely deranged vital signs on admission to an ICU in Malawi is a feasible prognostic marker for increased risk of in-hospital death—I think this conclusion needs further qualification due to the heterogeneity of the study population and the rate of missing data. Lines 331 to 339 provide an excellent summary as well as clear and helpful context. I think this is the core of the part of the discussion and would consider streamlining the rest of this section.

Thank you for these comments. We have qualified our conclusions in the discussion. And we agree that the discussion is long, however it is hard to streamline further than has already been done, as several parts were requested to be added during the review process. I hope that is ok. 

Study population heterogeneity: The authors state that “assessments of the performances of the severity scoring systems has limitations as the population in this study had different case-mix characteristics.” However, this also seems to be applicable to single deranged vital signs. I would consider acknowledging this and including discussion on how this limitation may have influenced the results.

Thanks, we have amended this to clarify that it concerns all results. 

b. Missing data: Please expand on the limitation that “there are some missing data”. There were significant rates of missing data, for the variables capillary refill time and Glasgow coma scale in particular. By providing the two sensitivity analyses in supplementary table 1 (i.e., case wise deletion of missing data and imputed abnormal data), the authors do an excellent job helping the reader to understand the potential impact of missing data on their estimates. I think the manuscript would benefit from a more detailed discussion in the text comparing estimates from the primary analysis with the two sensitivity analyses in table S1.

Thank you, this has been added. 

1) The NEWS score and any severely deranged vital sign had very similar point estimates in the primary analysis (Table 3). It therefore strikes me as inconsistent to say that “performances of the models were too low to be clinically useful by themselves for individual patient decisions” and to also suggest that severely deranged vital signs have clinical utility.

Thank you for this comment. We have clarified that we meant that all the scoring systems, including single severely deranged vital signs, do not have good enough performances to be used as sole markers for individual patient decisions, but that single severely deranged vital signs could be useful as a marker of an increased risk of death together with other clinical information. 

Minor comments

1. The authors state: “critical illness as defined by the binary cut-offs in each of the severity models showed a significant association with in-hospital mortality.” However, not all of the severity model binary cut-offs had a significant association with in-hospital mortality.

Thank you. The sentence has been amended. 

2. Tables 4 and 5: Percentages are missing from the first data column

Thank you, this has been added 

3. Supplementary table 1: What is the total number of patients with complete data included in these analyses?

Thank you for this observation. The total number of patients with complete data per each variable/factor has been added to the supplementary table 1. 

4. The manuscript needs additional copy editing.

The manuscript has been proof-read again by English language experts and changes made.

---

## [Decision Letter · Decision Letter 2]

12 Aug 2022

Factors associated with in-hospital mortality of patients admitted to an Intensive Care Unit in a tertiary hospital in Malawi

PONE-D-21-17302R2

Dear Dr. Mtisunge Kachingwe,

We’re pleased to inform you that your manuscript has been judged scientifically suitable for publication and will be formally accepted for publication once it meets all outstanding technical requirements.

Kind regards,

Vipa Thanachartwet, M.D.

Academic Editor

PLOS ONE

Additional Editor Comments (optional):

All issues were revised according to the reviewers' comments and suggestions.

Reviewers' comments:

Reviewer's Responses to Questions

**Comments to the Author**

1. If the authors have adequately addressed your comments raised in a previous round of review and you feel that this manuscript is now acceptable for publication, you may indicate that here to bypass the “Comments to the Author” section, enter your conflict of interest statement in the “Confidential to Editor” section, and submit your "Accept" recommendation.

Reviewer #1: All comments have been addressed

Reviewer #2: All comments have been addressed

2. Is the manuscript technically sound, and do the data support the conclusions?

Reviewer #1: Yes

Reviewer #2: Yes

3. Has the statistical analysis been performed appropriately and rigorously? 

Reviewer #1: Yes

Reviewer #2: Yes

4. Have the authors made all data underlying the findings in their manuscript fully available?

Reviewer #1: No

Reviewer #2: No

5. Is the manuscript presented in an intelligible fashion and written in standard English?

Reviewer #1: Yes

Reviewer #2: Yes

6. Review Comments to the Author

Reviewer #1: (No Response)

Reviewer #2: Dr. Kachingwe and colleagues have done an excellent job revising their manuscript--all comments are thoughtfully addressed.

The manuscript topic is of significant interest to the field, the data are well-presented, and the conclusions are well-reasoned and justified. I have no additional comments and congratulate the authors for their work.

7. PLOS authors have the option to publish the peer review history of their article (what does this mean?). If published, this will include your full peer review and any attached files.

Reviewer #1: No

Reviewer #2: **Yes: **Paul D Sonenthal, MD

---

## [Editor Report · Acceptance letter]

19 Aug 2022

PONE-D-21-17302R2 

Factors associated with in-hospital mortality of patients admitted to an Intensive Care Unit in a tertiary hospital in Malawi 

Dear Dr. Kachingwe:

I'm pleased to inform you that your manuscript has been deemed suitable for publication in PLOS ONE. Congratulations! Your manuscript is now with our production department. 

Kind regards, 

on behalf of

Associate Professor Vipa Thanachartwet 

Academic Editor

PLOS ONE